# Distribution, Biogeography and Characteristics of the Threatened and Data-Deficient Flora in the Southwest Australian Floristic Region

Carl R. Gosper *, Julia M. Percy-Bower, Margaret Byrne, Tanya M. Llorens and Colin J. Yates

Biodiversity and Conservation Science, Department of Biodiversity, Conservation and Attractions, Locked Bag 104, Bentley Delivery Centre, Kensington, WA 6983, Australia; julia.percy-bower@dbca.wa.gov.au (J.M.P.-B.); margaret.byrne@dbca.wa.gov.au (M.B.); tanya.llorens@dbca.wa.gov.au (T.M.L.); colin.yates@dbca.wa.gov.au (C.J.Y.)
* Correspondence: carl.gosper@dbca.wa.gov.au

**Abstract:** The Southwest Australian Floristic Region (SWAFR) supports an exceptional number of threatened and data-deficient flora. In this study, we: (i) collated statistics on the number, listing criteria and tenure of occurrence of threatened and data-deficient flora; (ii) conducted spatial and biogeographic analyses to address questions concerning patterns of diversity of threatened and data-deficient flora relative to the whole flora and evolutionary and threat drivers; and (iii) examined whether threatened and data-deficient flora richness is evenly distributed across plant lineages. We found that although threatened and data-deficient flora occurred across the breadth of the SWAFR, high richness was concentrated in a limited number of locations, which were not always strongly aligned with areas of higher land transformation. Data-deficient flora demonstrated different spatial patterns of occurrence to threatened flora. Approximately 70% of the populations of threatened and data-deficient flora occurred outside of lands managed primarily for conservation. Both evolutionary history and contemporary threats contribute to the current status and distribution of diversity of the threatened and data-deficient flora, with evolutionary history playing a significant role in predisposing a portion of the flora to having population traits that result in those flora meeting IUCN Red List criteria, along with ecological traits that predispose some to specific novel threats. An understanding of the distribution of species and threats, flora traits, and how these traits mediate susceptibility to threats, offers one potential way forward for an initial assessment of which of the 1819 data-deficient flora may be most at risk of extinction.

**Keywords:** biodiversity hotspot; evolutionary history; flora conservation; Mediterranean-type ecosystem; plant diversity; rarity; threatened species; threatening process

## 1. Introduction

The impact of threatening processes on plants has led to ~40% of vascular plant species globally being at risk of extinction [1,2]. However, spatial patterns of distribution of threatened (and recently extinct) flora are not uniform [1,3,4] due to the combined patterns of distribution of biodiversity and threats. Some regions, known as biodiversity hotspots [1,3,5], and specific locations within these regions [6–8], support particularly high concentrations of threatened and data-deficient species and populations. In addition to threatened species, ~8% of the global vascular flora is data deficient [2]. For these species, there is insufficient information for assessment of extinction risk [9]. An understanding of where threatened and data-deficient flora occurs and the species and population characteristics of the flora is essential for underpinning flora conservation [6–8,10].

Like other Mediterranean-climate regions, the Southwest Australian Floristic Region (SWAFR; as per Ref [11]; Figure 1) forms one of the world's biodiversity hotspots [5], supporting ~9355 native plant taxa. The flora of the SWAFR is characterised by high

levels of endemism, small geographic range sizes, edaphic specialisation, high alpha and beta diversity, relictual lineages, highly species-rich clades and trait specialisation [11–17]. Evolution on old, climatically buffered, infertile landscapes (OCBILs) has been proposed as contributing to many of the characteristics and traits of the SWAFR flora [18–21].

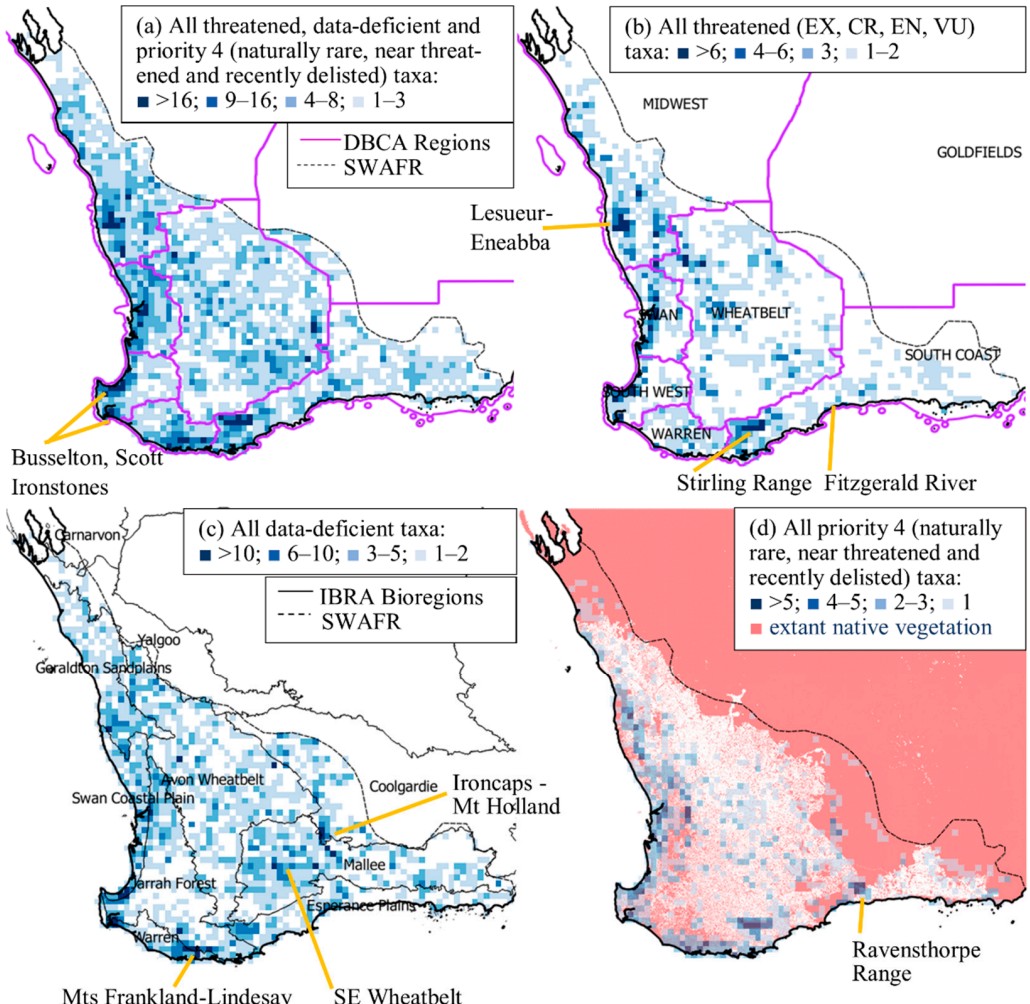

**Figure 1.** Distribution of taxa richness of different conservation statuses in the Southwest Australian Floristic Region (SWAFR): (**a**) all conservation-listed flora; (**b**) all threatened flora (extinct—EX, critically endangered—CR, endangered—EN, vulnerable—VU) in the context of Department of Biodiversity, Conservation and Attractions (BDCA) management regions; (**c**) all data-deficient flora in the context of Interim Biogeographic Regionalisation for Australia (IBRA) bioregions; and (**d**) all priority 4 (see Section 2.1) flora in the context of extant native vegetation. Locations referred to in the text are shown.

The SWAFR has high numbers of extinct, threatened and data-deficient taxa at state, national and global scales [1–3,22–24]. The sheer volume of threatened and data-deficient flora creates significant challenges for biodiversity conservation, particularly in combination with pervasive threats, including land clearance and associated habitat loss and fragmentation that increase the potential impacts of small population genetic and demographic processes, inappropriate fire regimes, secondary salinity, climate change, pathogen spread and invasive plants [21,22,25].

Early contributions to priority setting for conservation of the SWAFR's threatened and data-deficient flora focused on compiling the proportions of taxa affected by key threats and flora occurrence across land tenures and administrative regions [22,26,27]. These analyses provided significant insights into the relative magnitude and impact of threats but were

unable to explicitly consider detailed spatial patterns of distribution of threatened and data-deficient flora, nor apply these patterns in a modern spatially explicit conservation planning approach. A re-appraisal of the patterns of occurrence of threatened and data-deficient flora in the SWAFR, in a spatially explicit manner, is therefore highly beneficial.

Further, in the decades since Hopper et al. [26] and Coates and Atkins [22,27] published their findings, there has been a major focus on specimen acquisition and description of novel plant taxa, particularly those putatively of conservation concern due to exposure to mining activities [11,28]. This period has seen conservation activities being applied to threatened plant taxa and populations, such as expansion of the conservation estate, management of threats and conservation translocations [25]. Species have also been subject to continued threat pressure, emergence of new threats (or at least their recognition; e.g., senescence with long fire intervals [29–31]) and changed threat distribution (e.g., expanded mining on banded iron formation ranges [32]), and these changes in threats have affected the conservation status of some taxa. In addition, recent conceptual and empirical advances in understanding the evolution and function of the SWAFR flora as well as how evolutionary history influences susceptibility to threats [17,18,20,21,33] may support new interpretations of the patterns of threatened and data-deficient flora distribution and the management actions best suited to arrest decline and support flora recovery.

Prioritisation of conservation management actions may be improved through knowledge of the spatial patterns of distribution of threatened and data-deficient flora, how these patterns relate to evolution of flora diversity and plant traits, and how plant traits mediate susceptibility to threats [18,20,21,34]. Here, we calculate summary statistics on the listing criteria of threatened flora and land tenure of threatened and data-deficient flora populations and apply spatial and biogeographic analyses to explore the following questions:

1.   Are threatened and data-deficient flora evenly distributed across plant lineages and across the landscape?
2.   Do spatial and biogeographic patterns of threatened and data-deficient flora reflect those of the flora as a whole, or are they associated with evolutionary history or threats, such as macro-scale levels of land transformation, which is a threat commonly cited as an overwhelming contributor to extinction risk [35,36]?
3.   Are centres of high occurrence of data-deficient flora the same or a subset of those of the threatened flora, indicating that the conservation needs of data-deficient flora may at least partly be met by mitigating threats to natural populations of threatened flora?

We conclude that an understanding of the distribution of species and threats, the traits of flora and how these traits mediate susceptibility to threats offers one potential approach for an initial assessment of which of the 1819 data-deficient flora of the SWAFR may be most at risk of extinction.

## 2. Materials and Methods

### 2.1. Data Sources

Summary statistics concerning the whole flora of the SWAFR and the portion of threatened and data-deficient taxa were derived from a spatial intersect in a GIS between the database of the Western Australian Herbarium [37] and the SWAFR boundary [11] on 22 April 2022. The Western Australian Herbarium database is based on specimen collections and is the most comprehensive verifiable dataset of taxa occurrence. Limited spatial reliability of some records (e.g., pre-global positioning system) rendered this data source unsuitable for detailed spatial analysis; however, any spatial discrepancies in specimen location data are unlikely to be significant for the purpose of generating a species list at the scale of the SWAFR. All native taxa with a Western Australian Census name identifier were retained, which included phrase-name taxa and some hybrids but excluded taxa naturalised in the SWAFR.

Spatial analyses of threatened and data-deficient flora were conducted using a complete extract from the Department of Biodiversity, Conservation and Attractions (DBCA) Threatened and Priority Flora (TPFL) database on 22 March 2019 [38], intersected with

the boundary of the SWAFR in a GIS. The threatened flora (CR—critically endangered; EN—endangered; VU—vulnerable; and for the purposes of this study, including EX—extinct), as recognised by the State of Western Australia [39], were assessed according to the IUCN Red List criteria [9]. The priority flora list [22,26] is an additional list of taxa of conservation significance maintained by DBCA. Priority (P) 1–3 flora are essentially equivalent to data deficient under IUCN Red List criteria and are what we refer to when using the term 'data-deficient flora'. Priority 4 flora meet the adequacy of survey requirements for conservation status assessment and are those taxa that are naturally rare, near threatened or removed from the threatened species list in the last five years for reasons other than taxonomic change [39]. All entities included in TPFL were analysed at the level of a taxon; specifically, this included hybrids.

The TPFL database includes verified population locations, so records have high identification and spatial accuracy although with a lower quantum of records than specimen- or sighting-based databases. Like specimen-based data [11], TPFL data may have inherent sampling biases, which should be borne in mind when considering the results of spatial analyses. Specifically, it is plausible that population coverage is more comprehensive in densely settled parts of the SWAFR, transport corridors, known regions of high floristic richness and locations with more intense economic activity invoking legislative requirements for flora survey [11]. By definition, the comprehensiveness of population coverage in TPFL is lower for data-deficient than threatened flora. Flora status categories were aggregated in various combinations: (i) all conservation-listed (EX, CR, EN, VU, P1–4); (ii) threatened (EX, CR, EN, VU); (iii) data deficient (P1–3); and (iv) P4 (naturally rare, near threatened, recently delisted). Analyses were conducted including all flora per status category and for individual genera, focusing on a handful of genera that are species rich in threatened and data-deficient flora and iconic components of SWAFR biodiversity (*Eucalyptus* L'Her., *Acacia* Mill., *Caladenia* R.Br. and *Banksia* L.f.). Populations were defined as separate entries in the database, and fields in the database were used to define the tenure, land use and DBCA administrative region of all population occurrences in the SWAFR.

Both the Western Australian Herbarium and TPFL databases [37,38] include vascular and non-vascular flora, lichens and fungi. As lichens, fungi and non-vascular flora form only a very small proportion of records in these databases, the patterns described herein are largely attributable to the vascular flora.

*2.2. Data Analyses*

BIODIVERSE [40] software (Version 2.1, https://github.com/shawnlaffan/biodiverse; accessed on 29 March 2019) was used to aggregate the number of populations and richness of taxa of threatened, data-deficient and priority 4 flora from the TPFL database in cells of 0.125° latitude and longitude (ca. 14 × 12 km at 32° S). Endemism (using the endemism whole, weighted endemism metric) considered the adjoining four cells as the neighbour set, and the agglomerative cluster analysis used the Sorenson dissimilarity metric with the link average cluster linkage method, with cells with three or fewer taxa or five or fewer populations omitted.

## 3. Results

*3.1. Characteristics of the Conservation-Listed Flora*

The SWAFR has 406 taxa of threatened flora (~4.3% of the total flora), 1819 data-deficient flora (~19.5%) and 325 priority 4 flora (~3.5%) (Table 1). Many of the genera with the greatest number of threatened flora are among the most species rich in the SWAFR, with the proportion of threatened taxa being little different to the overall SWAFR average (e.g., *Acacia*, *Eucalyptus*; Table 1).

**Table 1.** Genera with notably high proportional representation of threatened and data-deficient taxa, threatened taxa and high raw numbers of threatened taxa, relative to the whole flora of the Southwest Australian Floristic Region (SWAFR).

| Genus | Number of SWAFR Taxa | Number of Threatened (T) Taxa | % Genus T | Number of Data-Deficient (DD) Taxa | % Genus DD | % Genus T + DD |
|---|---|---|---|---|---|---|
| High proportional representation [1] (T + DD) | | | | | | |
| *Drummondita* | 7 | 2 | 28.6 [#] | 3 | 42.9 | 71.4 |
| *Brachyloma* | 10 | 0 | 0 | 7 | 70.0 | 70.0 |
| *Babingtonia* | 12 | 0 | 0 | 8 | 66.7 | 66.7 |
| *Netrostylis* | 6 | 0 | 0 | 4 | 66.7 | 66.7 |
| *Calectasia* | 14 | 2 | 14.3 [#] | 7 | 50 | 64.3 |
| *Cyathostemon* | 11 | 0 | 0 | 7 | 63.6 | 63.6 |
| *Malleostemon* | 16 | 0 | 0 | 10 | 62.5 | 62.5 |
| *Eryngium* | 8 | 0 | 0 | 5 | 62.5 | 62.5 |
| *Enekbatus* | 8 | 0 | 0 | 5 | 62.5 | 62.5 |
| *Synaphea* | 76 | 6 | 7.9 | 40 | 52.6 | 60.5 |
| High proportional representation [1] (T only) | | | | | | |
| *Drakaea* | 9 | 5 | 55.6 | 0 | 0 | 55.6 |
| *Lambertia* | 19 | 5 | 26.3 | 1 | 5.3 | 31.6 |
| *Darwinia* | 72 | 16 [^] | 22.2 | 19 | 26.4 | 48.6 |
| *Androcalva* | 14 | 3 | 21.4 | 3 | 21.4 | 42.9 |
| *Myoporum* | 10 | 2 | 20.0 | 0 | 0 | 20.0 |
| *Commersonia* | 12 | 2 | 16.7 | 1 | 8.3 | 25.0 |
| *Trithuria* | 6 | 1 | 16.7 | 0 | 0 | 16.7 |
| *Tetratheca* | 35 | 5 | 14.3 | 15 | 42.9 | 57.1 |
| *Eremophila* | 133 | 19 [^] | 14.3 | 22 | 16.5 | 30.8 |
| *Acrotriche* | 7 | 1 | 14.3 | 0 | 0 | 14.3 |
| High raw numbers [1] (T) | | | | | | |
| *Acacia* | 586 | 32 | 5.5 | 131 | 22.4 | 27.8 |
| *Grevillea* | 288 | 30 | 10.4 | 74 | 25.7 | 36.1 |
| *Eucalyptus* | 431 | 26 | 6.0 | 42 | 9.7 | 15.8 |
| *Caladenia* | 196 | 25 | 12.8 | 26 | 13.3 | 26.0 |
| *Banksia* | 237 | 20 | 8.4 | 45 | 19.0 | 27.4 |
| *Verticordia* | 161 | 16 | 9.9 | 46 | 28.6 | 38.5 |
| *Daviesia* | 112 | 12 | 10.7 | 15 | 13.4 | 24.1 |
| *Gastrolobium* | 113 | 12 | 10.6 | 27 | 23.9 | 34.5 |
| *Conostylis* | 80 | 9 | 11.3 | 6 | 7.5 | 18.8 |
| *Stylidium* | 233 | 7 | 3.0 | 67 | 28.8 | 31.8 |
| SWAFR total | 9355 | 406 | 4.3 | 1819 | 19.4 | 23.8 |

[1] For genera with >5 taxa. Genera also in the top 10 genera for high proportional representation of threatened taxa ([#]) or with high raw numbers of threatened taxa ([^]). Many genera had no threatened or data-deficient taxa.

However, there is strong evidence that other genera and families are either strongly over-represented (e.g., the genera *Drakaea* Lindl., *Lambertia* Sm. and *Darwinia* Rudge have >5 times the proportion of threatened flora compared to the SWAFR average, whereas the families Scrophulariaceae, Elaeocarpaceae and Dasypogonaceae have >2 times the average) or under-represented (e.g., *Amyema* Tiegh. (Loranthaceae), *Callitris* Vent. (Cupressaceae) and *Cassytha* Osbeck (Lauraceae) have no threatened or data-deficient taxa) in the conservation-listed flora (Table 2). Stark differences in representation in the threatened flora occur even between genera within the same plant family, for example, *Grevillea* Knight (10.4% of taxa threatened) and *Hakea* Schrad. and J.C.Wendl. (1.7%) in Proteaceae, and *Darwinia* (22.2%) and *Melaleuca* L. (1.0%) in Myrtaceae.

Threatened flora are most frequently listed under IUCN criterion B (geographic range in the form of either extent of occurrence and/or area of occupancy, with fragmentation or few locations, and decline), followed by D (very small or restricted population), C (small population size and decline) and A (population size reduction)—with none currently listed under criterion E (Quantitative Analysis) (Table 3). Many taxa are listed under multiple criteria. However, for those taxa listed under a single criterion, criterion D appears particularly strongly represented relative to total listings under that criterion.

**Table 2.** Plant families with notably high or low proportional representation of threatened and data-deficient taxa and high representation of threatened taxa, relative to the whole flora of the Southwest Australian Floristic Region (SWAFR).

| Family | Number of SWAFR Taxa | Number of Threatened (T) Taxa | % Family T | Number of Data-Deficient (DD) Taxa | % Family DD | % Family T + DD |
|---|---|---|---|---|---|---|
| High proportional representation [1] (T + DD) | | | | | | |
| Elaeocarpaceae | 20 | 5 | 11.9 [#] | 17 | 40.5 | 52.4 |
| Dasypogonaceae | 19 | 2 | 10.5 [#] | 7 | 36.8 | 47.4 |
| Araliaceae | 41 | 0 | 0 | 14 | 34.1 | 34.1 |
| Gyrostemonaceae | 18 | 1 | 5.6 | 5 | 27.8 | 33.3 |
| Hemerocallidaceae | 60 | 1 | 1.7 | 19 | 31.7 | 33.3 |
| Malvaceae | 202 | 14 | 6.9 [#] | 52 | 25.7 | 32.7 |
| Rhamnaceae | 123 | 0 | 0 | 40 | 32.5 | 32.5 |
| Brassicaceae | 38 | 1 | 2.6 | 11 | 28.9 | 31.6 |
| Stylidiaceae | 243 | 7 | 2.9 | 69 | 28.4 | 31.3 |
| Ericaceae | 398 | 12 | 3.0 | 108 | 27.1 | 30.2 |
| High proportional representation [1] (T only) | | | | | | |
| Scrophulariaceae | 147 | 21 | 14.3 | 22 | 15.0 | 29.3 |
| Haemodoraceae | 137 | 12 | 8.7 | 12 | 8.7 | 17.5 |
| Frankeniaceae | 23 | 2 | 8.7 | 4 | 17.4 | 26.1 |
| Casuarinaceae | 35 | 3 | 8.6 | 4 | 11.4 | 20.0 |
| Orchidaceae | 494 | 42 | 8.5 | 61 | 12.3 | 20.9 |
| Xyridaceae | 12 | 1 | 8.3 | 1 | 8.3 | 16.7 |
| Polygonaceae | 13 | 1 | 7.7 | 0 | 0.0 | 7.7 |
| Proteaceae | 1032 | 77 | 7.5 | 216 | 20.9 | 28.4 |
| Colchicaceae | 28 | 2 | 7.1 | 1 | 3.6 | 10.7 |
| Rutaceae | 206 | 12 | 5.8 | 41 | 19.9 | 25.7 |
| Low proportional representation [1] (T + DD) | | | | | | |
| Cupressaceae | 13 | 0 | 0 | 0 | 0 | 0 |
| Lauraceae | 16 | 0 | 0 | 0 | 0 | 0 |
| Loranthaceae | 17 | 0 | 0 | 0 | 0 | 0 |
| Potamogetonaceae | 14 | 0 | 0 | 0 | 0 | 0 |
| Ranunculaceae | 13 | 0 | 0 | 0 | 0 | 0 |
| Zygophyllaceae | 21 | 0 | 0 | 0 | 0 | 0 |
| Xanthorrhoeaceae | 11 | 0 | 0 | 0 | 0 | 0 |
| Campanulaceae | 33 | 0 | 0 | 1 | 3.0 | 3.0 |
| Crassulaceae | 19 | 0 | 0 | 1 | 5.3 | 5.3 |
| Convolvulaceae | 17 | 0 | 0 | 1 | 5.9 | 5.9 |
| SWAFR total | 9355 | 406 | 4.3 | 1819 | 19.4 | 23.8 |

[1] For families with >10 taxa. [#] Families also in the top 10 families for high proportional representation of threatened taxa.

**Table 3.** IUCN Red List criteria under which the threatened flora of the Southwest Australian Floristic Region are listed under state legislation.

| Criterion | Solely Listed under Criterion | Listed under Criterion and Other Criteria [1] | Total |
|---|---|---|---|
| Total *A* (Population size reduction) | 11 | 24 | 35 |
| *A1* (past; reversible, understood, ceased) | 0 | 7 | 7 |
| *A2* (past; not reversible, not understood or not ceased) | 5 | 8 | 13 |
| *A3* (projected) | 5 | 5 | 10 |
| *A4* (past and projected; not reversible, not understood or not ceased) | 1 | 5 | 6 |
| Total *B* (Geographic range and fragmentation, number of locations, decline and/or fluctuation) | 127 | 107 | 234 |
| *B1* (Extent of occurrence) *ab* (locations or fragmentation + continuing decline) | 8 | 148 | 156 |
| *B2* (Area of occupancy) *ab* (locations or fragmentation + continuing decline) | 18 | 153 | 171 |
| Other *B1*, *B2* combinations | 29 | 20 | 49 |
| Total *C* (Small population size and decline) | 38 | 92 | 130 |
| *C1* (continuing decline) | 2 | 21 | 23 |
| *C2* (continuing decline, subpopulation, population fluctuation) | 35 | 79 | 114 |
| Total *D* (Very small or restricted population) | 100 | 57 | 157 |
| *D* (Mature individuals) | 61 | 58 | 119 |
| *D2* (VU; restricted + plausible future threat) | 36 | 7 | 43 |
| Total *E* (Quantitative Analysis) | 0 | 0 | 0 |

[1] Taxa can be listed under multiple criteria. One taxon with incomplete listing criteria [39] was not included.

### 3.2. Spatial Distribution of Conservation-Listed Flora

Threatened, data-deficient and priority 4 (naturally rare, near threatened and recently delisted) taxa richness and population density (both exhibit similar spatial patterns) are disproportionately distributed in a limited number of specific locations, although a low density occurs across much of the SWAFR (Figure 1). Strikingly, many of the landscapes with concentrations of threatened and data-deficient flora have high cover of extant native vegetation (Figure 1d).

There were distinct differences in patterns of distribution between threatened, data-deficient and priority 4 flora. Threatened flora occurrence was greatest in near-coastal locations in the western and southern SWAFR (Figure 1b). While data-deficient flora is strongly represented in most of the hotspots of threatened flora, an additional set of locations stands out as having uniquely high occurrence of data-deficient flora, particularly in the southeast inland and southern near-coastal areas (Figure 1c). Priority 4 flora are even more strongly concentrated than the threatened flora around the relatively mesic coastal and sub-coastal periphery of the SWAFR (Figure 1d). Individual genera also demonstrated distinct patterns of distribution (Figure 2).

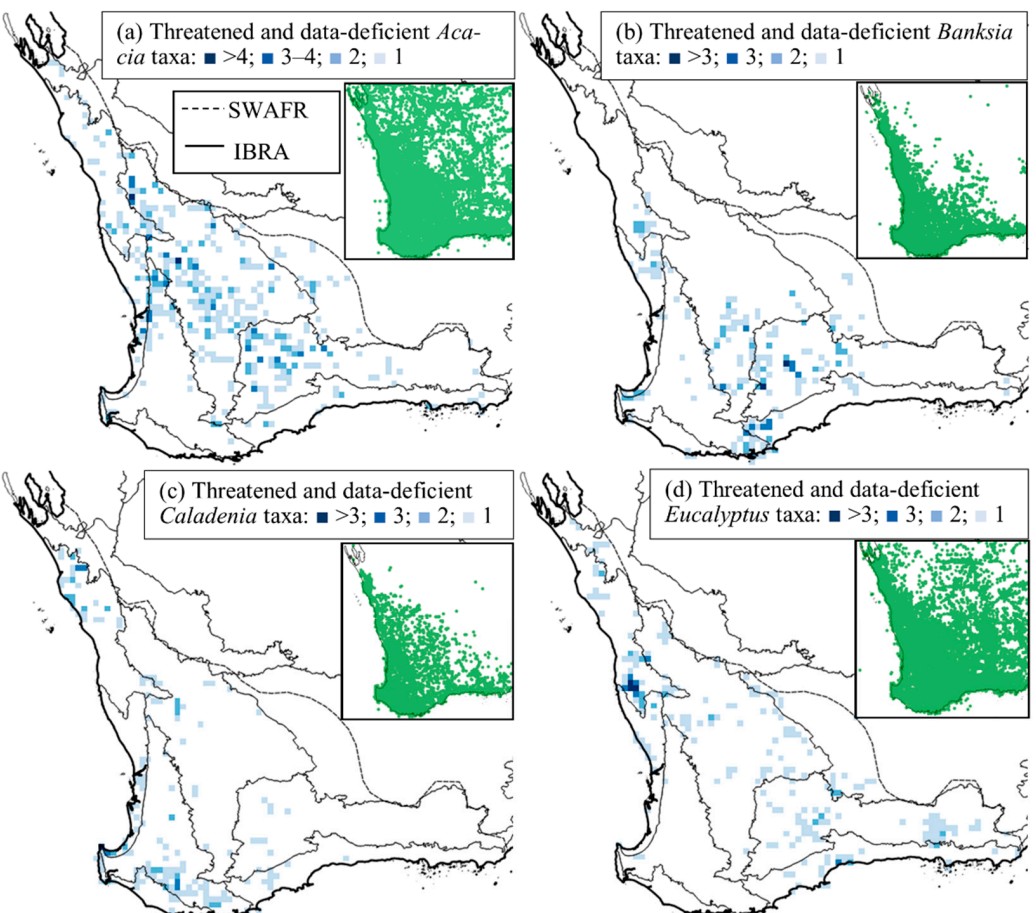

**Figure 2.** Distribution of taxa richness in selected major individual genera of threatened and data-deficient flora in the context of the Southwest Australian Floristic Region (SWAFR) and Interim Biogeographic Regionalisation for Australia (IBRA) bioregions: (**a**) *Acacia*; (**b**) *Banksia*; (**c**) *Caladenia*; and (**d**) *Eucalyptus*. Inserts show distribution of all records of these genera held in the Atlas of Living Australia (https://www.ala.org.au/) on 2 May 2022.

### 3.3. Administrative Regions and Tenure of Conservation-Listed Flora

The DBCA administrative regions with the highest numbers of threatened flora taxa and populations are Wheatbelt, Midwest and South Coast (Table 4). Across the SWAFR, the single land tenure type supporting the largest number of populations of threatened, data-

deficient and priority 4 flora is DBCA-managed conservation estate (Figure 3a). However, 69% of populations occur outside of the conservation estate, with private freehold land and road verges managed by local government authorities supporting a high proportion of populations. The distribution of populations across tenures differs dramatically across the SWAFR. In the Warren DBCA region, which has relatively high natural vegetation cover and large areas of state-managed land, nearly two-thirds of threatened, data-deficient and priority 4 flora populations occur on DBCA-managed conservation estate, with another ~10% in state forests managed by the state for a range of public benefits (Figure 3b). In contrast, in the Wheatbelt DBCA region, where large areas of native vegetation have been cleared, only 23% of populations occur on the conservation estate. High proportions of populations occur on road verges, private land, unallocated crown land and non-conservation state government tenure (Figure 3c) where much of the remaining native vegetation in the agricultural landscape occurs.

**Table 4.** Number and distribution of threatened taxa and populations among Department of Biodiversity, Conservation and Attractions (DBCA) administrative regions (Figure 1b) in the Southwest Australian Floristic Region (SWAFR). Note that: (i) a small number of extinct (EX) flora present in the Herbarium specimen database (Table 1) have no records in the Threatened and Priority Flora database used in generating these figures; and (ii) only part of South Coast, Wheatbelt, Goldfields and Midwest regions overlaps the SWAFR.

| Status | DBCA Region | | | | | | | Total |
|---|---|---|---|---|---|---|---|---|
| | South Coast | Warren | South West | Wheatbelt | Swan | Goldfields | Midwest | |
| Taxa | | | | | | | | |
| EX | 0 | 0 | 2 | 1 | 0 | 0 | 0 | 3 [1] |
| CR | 36 | 7 | 26 | 60 | 24 | 0 | 44 | 153 [1] |
| EN | 32 | 12 | 22 | 48 | 27 | 0 | 44 | 131 [1] |
| VU | 45 | 12 | 13 | 53 | 30 | 2 | 31 | 116 [1] |
| Total | 113 | 31 | 63 | 162 | 81 | 2 | 119 | 403 |
| Populations | | | | | | | | |
| EX | 0 | 0 | 2 | 7 | 0 | 0 | 0 | 9 |
| CR | 393 | 23 | 183 | 362 | 279 | 0 | 340 | 1580 |
| EN | 301 | 118 | 289 | 549 | 258 | 0 | 521 | 2036 |
| VU | 375 | 152 | 109 | 555 | 426 | 5 | 400 | 2022 |
| Total | 1069 | 293 | 583 | 1473 | 963 | 5 | 1261 | 5647 |

[1] Total refers to the number of taxa per status category, not the sum across regions, as some taxa occur in multiple regions.

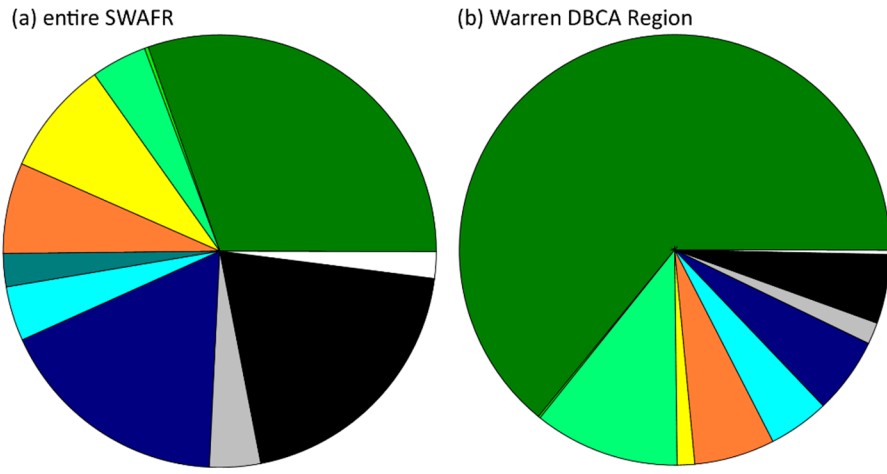

(a) entire SWAFR    (b) Warren DBCA Region

**Figure 3.** *Cont.*

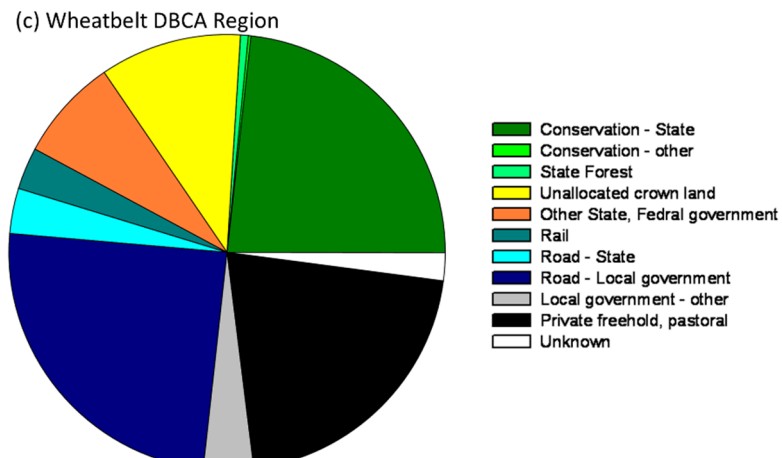

**Figure 3.** Tenure and land use of all threatened, data-deficient and priority 4 flora populations in: (**a**) the Southwest Australian Floristic Region (SWAFR); (**b**) Warren; and (**c**) Wheatbelt Department of Biodiversity, Conservation and Attractions (DBCA) management regions.

### 3.4. Endemism and Biogeographic Patterns of Conservation-Listed Flora

Endemism among the threatened, data-deficient and priority 4 flora of the SWAFR at the 0.125° scale showed similar patterns of areas of high and low endemism to areas of high and low numbers of taxa (Figure 4a cf. 1). The phytogeography analysis showed an initial branch of the dendrogram separated the flora of the coastal north (Group 1; northern Geraldton Sandplains and coastal Yalgoo bioregions), followed by Groups 2–5 extending across the most mesic parts of the SWAFR from the southern Geraldton Sandplains bioregion through the Swan Coastal Plain, Jarrah Forest and Warren, to the western Esperance Plains (Figure 4d). The last-branching groups occurred in the southeast coast (Group 6), followed by the remaining Groups 7–10 in the more xeric interior of the SWAFR.

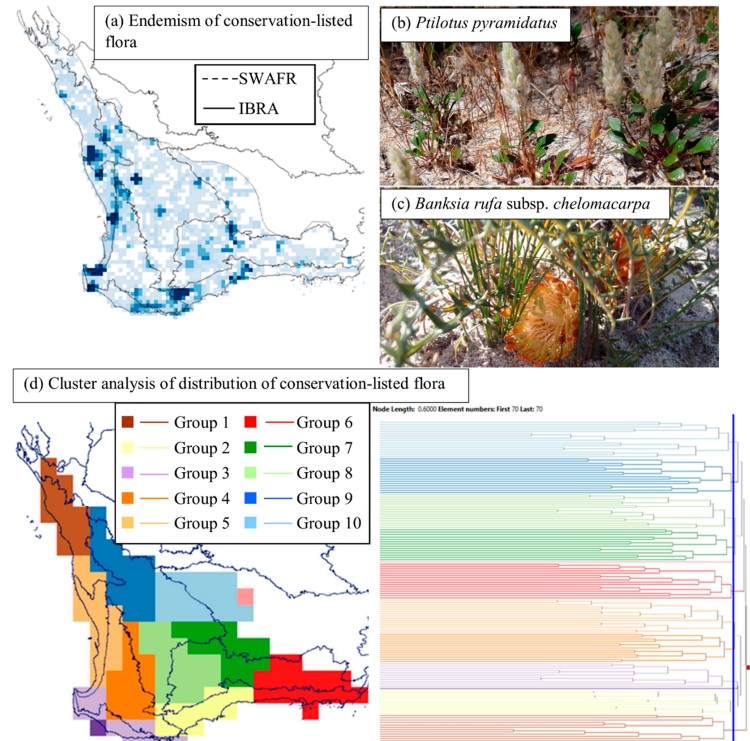

**Figure 4.** Biogeography of threatened, data-deficient and priority 4 flora in the Southwest Australian Floristic Region (SWAFR): (**a**) endemism at the scale of 0.125° cells scaled into five equal-sized colour

ramp classes (light to dark); examples of (**b**) threatened and (**c**) data-deficient flora (photos C. Gosper); and (**d**) an agglomerative cluster analysis of composition based on 0.5° cells. The spatial distribution of clusters (**left**) matches the colour of branches of the dendrogram (**right**), with the thick vertical blue line of the dendrogram showing the level of dissimilarity at which clusters were separated.

## 4. Discussion

### 4.1. Biogeographic Patterns—Hotspots for Conservation-Listed Flora

The spatial pattern of occurrence of threatened, data-deficient and priority 4 flora in the SWAFR, as a whole or in separate status categories, was neither uniform nor random. The concentration of threatened and data-deficient flora taxa and populations in specific parts of the landscape is typical of regional-scale assessments globally [6–8]. The greatest concentrations of conservation-listed flora occurrence in the SWAFR were on specific, usually ancient, geomorphological features, not widespread across the broader landscape (Figure 1; [20]). A low density of conservation-listed flora occurred across highly cleared agricultural and urban landscapes of the Wheatbelt (particularly data-deficient taxa) and Swan Coastal Plain, respectively, and these highly transformed landscapes contained some hotspots of conservation-listed flora. However, several of the geomorphological features supporting many conservation-listed flora retained high cover of native vegetation, illustrating a decoupling of conservation-listed flora occurrence from macro-scale levels of land transformation ([20] cf. [35,36]).

The characteristics and evolutionary history of the geomorphological features supporting conservation-listed flora have a large bearing on the ecological, population and genetic traits of the flora and the prevailing threats to which the flora are exposed [21]. As specific plant traits have an important role in mediating susceptibility to threats [18,34,41], it follows that for differing trait profiles and pervasive threats at each hotspot of threatened, data-deficient and naturally rare flora occurrence, conservation will be best served by individually tailored management approaches.

The Wheatbelt, Midwest and South Coast DBCA Regions contained the largest numbers of threatened flora taxa and populations, which is a pattern that has remained consistent over time [22,26], despite increases in the numbers of threatened taxa, known populations and changes in the understanding and distribution of threats [21,22,42]. Although the quantity of known threatened flora populations and the number of populations on the conservation estate have grown over time with new population discoveries, new listings and additions to the conservation estate, the proportion of threatened flora populations known from the conservation estate relative to other land tenures and land uses has remained fairly static at ~30% (Figure 3, cf. [27]).

Among individual genera, there was sometimes broad consistency between areas of high richness of threatened and data-deficient taxa within the genus and the patterns of species richness in the genus overall (*Acacia* cf. [17], *Banksia* cf. [43]). However, spatial patterns of occurrence of threatened and data-deficient *Eucalyptus* and *Caladenia* differ starkly from areas of greatest species richness in those genera overall. *Eucalyptus* species richness is greatest towards the southeastern coast [17,36]; yet, threatened and data-deficient *Eucalyptus* occurrence was greatest much further north in the Lesueur-Eneabba area. Although the reasons for these contrasting spatial patterns are not clear, intriguingly, several of the threatened *Eucalyptus* in the Lesueur-Eneabba area are hybrids [37]. *Caladenia* species richness peaks in the relatively mesic southwest [44], whereas threatened and data-deficient *Caladenia* have a pronounced concentration in the relatively arid far north, where, overall, few *Caladenia* occur. Climate has been proposed as a driver of rarity in *Caladenia* [44].

The contrasting proportions of threatened flora among families and genera are unable to be explained by differences in distribution mediating differential exposure to land transformation. *Banksia*, *Melaleuca* and *Eucalyptus* each have broadly similar spatial patterns of distribution of species richness, with peak richness towards the southeastern coast of the SWAFR [17,36,43]. This part of the SWAFR does not have the extreme levels of land transformation of other parts of the SWAFR (Figure 1); yet, *Banksia* has relatively high proportions of threatened flora, while the other genera do not, indicating differential

susceptibility to other threats. In the case of *Banksia*, the high proportion of conservation-listed flora appears to be related to a combination of critical threats [45], namely high susceptibility to *Phytophthora* dieback [46] along with vulnerability to short fire intervals via evolution of (locally predominant) an obligate-seeder fire response strategy with a long juvenile period associated with serotiny and low-productivity environments [47]. In contrast, *Eucalyptus* and *Melaleuca* do not show high richness of conservation-listed flora on the southeastern coast, likely because they are more resilient to *Phytophthora* infection (they lack the cluster root specialisation for phosphorus acquisition of *Banksia*, which is associated with greater disease susceptibility [46,48]) and rarely feature as the slowest-maturing obligate-seeder species [47].

Endemism among the threatened, data-deficient and priority 4 flora of the SWAFR at the 0.125° scale showed similar patterns to the number of taxa (Figure 4a, cf. 1a). The essentially similar spatial patterns of richness and endemism indicate that very few of these taxa have geographic ranges extending between hotspots, consistent with the highly restricted range sizes of threatened and data-deficient flora [20].

The overall pattern of richness and endemism of threatened, data-deficient and priority 4 flora in the SWAFR shares many similarities to that of the flora of the region as a whole [11]. The Lesueur-Eneabba uplands, the quartzite Stirling Range, greenstone Ravensthorpe Range and eastern Swan Coastal Plain are all exceptionally species rich and have high levels of endemism for both conservation-listed and the whole flora. This pattern of diversity emphasises the importance of restricted geomorphological features, usually of great antiquity and/or with highly impoverished soils, in driving evolution of a flora characterised by narrow-range endemics [14,18,20], even though non-conservation-listed flora have larger geographic ranges than threatened flora [20]. Further, the dominance of flora being listed under IUCN Red List criteria B, C and D (cf. A) also suggests that the bulk of the recognised threatened flora have relatively narrow geographic ranges and/or small population sizes. The dominance of listings under criterion B in the SWAFR is consistent with global patterns, but the frequency of listings under criteria C and D is much greater than the global average but similar to Australia as a whole [10]. The greater data requirements for supporting listings under criterion A may also contribute to the low numbers of SWAFR taxa listed under this criterion. Thus, rarity and narrow-range endemism appear to have played a strong role preceding European colonisation in driving the composition and spatial patterns of distribution of contemporary SWAFR conservation-listed flora.

However, within the subset of the SWAFR with particularly high richness and high levels of narrow-range endemism in the flora, threat intensity appears to have an important role in determining the relative numbers of threatened and data-deficient flora. For example, the Stirling Range, where the flora is highly threatened by the combination of *Phytophthora* disease and short fire intervals [45], has a much greater concentration of threatened and data-deficient flora than the Ravensthorpe Range-Fitzgerald River National Park area (Figure 1b) where threats have been less pronounced to date. This is despite both areas supporting similarly high floristic richness and endemism [11], frequent shared taxa and similarities in geomorphology and landscape evolutionary history [20].

Biogeographic patterns of distribution of conservation-listed flora are also very similar to those of the whole flora (Figure 4b; [11]). We interpret this as demonstrating that conservation-listed flora and the flora as a whole exhibit similar patterns of frequent local endemism on restricted, old and unusual geomorphological features [12,14,18,20], combined with high spatial turnover on many of the more widespread land surfaces [15,16]. Restricted ranges and small population sizes, with or without evidence of decline, can contribute to taxa meeting the IUCN Red List criteria [9].

### 4.2. The Elephant in the Room—The Data-Deficient Flora

The large quantum of data-deficient flora in the SWAFR is a substantial constraint on effective conservation management. Undescribed (informally named) taxa are over-

represented among the data-deficient flora [42], illustrating an additional taxonomic impediment to effective conservation. For all data-deficient flora, there is insufficient information to robustly assess the conservation status [9]. Thus, it is unknown whether: (i) these taxa are not at high risk of extinction, such that further surveys would reveal them to be more widespread; or (ii) whether inadequate survey data obscure species at high extinction risk. Most of the locations with high numbers of threatened flora also support many data-deficient flora. However, there are several additional locations for which the bulk of the conservation-listed taxa are data deficient, suggesting that conservation activities directed at mitigating threats to natural populations of threatened flora will not be effective for the conservation management of a significant proportion of the data-deficient flora. Hotspots solely important for data-deficient flora include sandplains of the southeastern Wheatbelt, the Parker and Ironcaps-Mt Holland greenstone ranges and the granites of the Mts Frankland-Lindesay area. Although the spatial analyses of patterns of occurrence of data-deficient flora are likely limited by inadequate data, similar concentrations of population occurrence in particular locations are also apparent in Brazil [8].

Surveys to quantify the conservation status of data-deficient flora are clearly desirable and are the optimal approach to overcoming uncertainty over the true status. While progress is being made [49], it is a significant undertaking, given the 1819 data-deficient flora in the SWAFR. Indeed, the trend over multiple decades is for increasing rather than decreasing numbers of data-deficient flora [22], in line with ongoing discovery of new taxa from the SWAFR [11,42]. Approaches for estimating the extinction risk in the absence of adequate population data would assist in effective management of these species.

A combination of a sampling effort and trait- and threat-based approach offers one means by which the multitude of data-deficient flora could be prioritized into those with a higher likelihood of truly being at risk of extinction and those at lower risk. Alternatively, approaches based on assessing the distribution data against IUCN Red List criterion B have also shown promise [10,50]. Notably, with the sole exception of the Mts Frankland-Lindesay area, all of the aforementioned hotspots for data-deficient (but not threatened) flora have moderate to low sampling density in the Western Australian Herbarium database relative to the remainder of the SWAFR [11], suggesting potential sampling artefacts. However, many of these landscapes have a very high intensity of land use and high levels of threat. In contrast, all locations with high concentrations of threatened flora are strongly represented in the herbarium collections.

It follows, in the absence of additional survey data, that data-deficient flora occurring in well-collected parts of the SWAFR (e.g., Lesueur-Eneabba, eastern Swan Coastal Plain-Darling Scarp, Busselton and Scott River ironstones, Stirling Range and Mts Frankland-Lindesay) are indeed likely to be restricted in distribution unless there are clear reasons why flora detectability and collectability may be low. Potential reasons for low representation in collections may include taxonomic confusion [42] or ecological attributes that limit detectability, such as a short-lived post-fire ephemeral disturbance strategy in infrequently burnt ecosystems or low and/or cryptic plant stature.

The value of using traits and threats in the initial assessment of extinction risk in data-deficient flora can be illustrated by examples in three regions where there is a concentration of data-deficient flora: the Stirling Range, southeastern Wheatbelt and Ironcaps-Mt Holland range. Most of the land area in the Stirling Range remains covered by native vegetation and is managed for conservation by DBCA, suggesting that risks of extensive land clearance and fragmentation are low. The combination of *Phytophthora* infestation and short fire intervals is implicated in numerous population-level extinctions in the Stirling Range [45]. Only a subset of the flora is susceptible to these threats, with some plant families showing much greater susceptibility than others to *Phytophthora* [46] and obligate-seeder species with relatively long juvenile periods, particularly those with canopy seed banks, to short fire intervals [45,47]. Data-deficient flora from the Stirling Range with one or both of these traits (either known or inferred from related species) would seem at more risk of extinction than data-deficient flora lacking these traits. Examples of data-deficient species that may

meet these criteria based on congeneric responses [46] include several *Andersonia* R.Br., *Gastrolobium leakeanum* J.Drumm. and *Daviesia mesophylla* Ewart.

The southeastern Wheatbelt has experienced extensive land clearance for agriculture, with much of the remaining native vegetation restricted to a small number of large reserves and many small reserves and linear remnants of vegetation along roadsides [21,51]. Although past land clearance is highly significant, future land clearance on broad scales in this region is unlikely [52]. The primary threats to flora are senescence with long fire intervals associated with disruption of spatial patterns of ignition and fuel continuity, fragmentation and associated processes, such as small population size and edge effects, weed invasion at nutrient-enriched remnant edges and secondary salinity in lower parts of the landscape [21,25,30,51,53,54]. All data-deficient flora known predominantly from lower landscape positions are presumably at high risk from secondary salinity, irrespective of other traits. The fire response and seed bank traits of data-deficient flora from the uplands will mediate susceptibility to senescence risks. Obligate seeders with persistent soil-stored seed banks, including the post-fire ephemeral subset of species with limited longevity, have the capacity to persist as viable propagules in the soil after the disappearance of above-ground plants [25], offering the potential for 'recovery' of populations with no extant plants and discovery of 'new' populations after disturbances such as fire. Obligate seeders with canopy-stored seed banks lack this mechanism of population persistence after loss of adult plants [30,47], so they would appear to be at higher risk of extinction in the southeastern Wheatbelt given the prevailing threats.

The Ironcaps-Mt Holland area lies towards the xeric margin of the SWAFR, so it has not been substantially impacted by land clearance for agriculture. However, the mafic igneous (greenstone) and banded iron features of the low ranges of this area are prospective for a variety of minerals, and impacts from mining exploration and mining operations are widespread [55]. The extinction risk of the data-deficient flora of this part of the SWAFR is likely to be strongly linked to the geomorphological habitat preferences of each taxon. Those data-deficient flora that are restricted to greenstone and ironstone geologies, and which constitute the bulk of the narrow-range endemics of the area [32,56], are likely to be at higher extinction risk. Data-deficient flora occurring on widespread (non-greenstone and non-ironstone) geologies, or across a broad range of geologies, would seem to be at lower risk, although still exposed to some mining footprint impacts [55] and other threats, such as short fire intervals (in the case of obligate seeders; [30,57]). As much of the surrounding area is remote from roads and has relatively low plant collection effort [11], it is plausible that substantial undiscovered populations of data-deficient flora occurring on widespread geologies may exist.

The richness of priority 4 flora (naturally rare, near threatened and recently delisted) was strongly concentrated in a number of specific locations in the more mesic coastal and sub-coastal parts of the SWAFR. This pattern of distribution is consistent with high flora collection effort in coastal parts of the SWAFR [11], such that there are sufficient data to support conservation status assessment. While persistence in localized mesic refugia following ongoing aridification over geological time [58] contributes to small populations in much of the SWAFR, the southern coastal area has also been shown to be a location of a major mesic refugium in a widespread species [59] and may be an explanation of the increased density of naturally rare species if those species contracted to this refugium but were then unable to expand their distribution in more mesic times. Understanding the spatial patterns of distribution of priority 4 flora in combination with susceptibility to threats allows for a more rapid response to the emergence of new threats or intensification of existing threats. For example, the Ravensthorpe Range supports many priority 4 flora (Figure 1d; [60]) but not unusually high richness of threatened or data-deficient flora. Expanded mining operations [60] could result in some of these priority 4 taxa meeting the criteria for being listed as threatened. For those priority 4 flora in mesic refugia, ongoing climatic warming and drying poses a particularly significant threat [21,61]. The intensification of the threat of *Phytophthora* infestation in the Stirling Range magnified by

past short fire intervals [45] suggests that ongoing monitoring of priority 4 flora with either or both *Phytophthora* susceptibility and an obligate-seeder fire response strategy would inform population dynamics and conservation actions. Several priority 4 *Banksia* spp., for example, are wholly endemic to the Stirling Range.

## 5. Conclusions

Spatial analysis of the distribution of threatened and data-deficient flora across the Southwest Australian Floristic Region showed that populations were concentrated in particular locations, not always coinciding with areas of greatest land transformation. Hotspots for threatened flora only formed a subset of hotspots for data-deficient flora, indicating that threat mitigation actions for threatened flora will not meet the conservation needs of all data-deficient flora. Lineages of plant species were not evenly represented among the threatened and data-deficient flora; this finding, in combination with patterns of spatial distribution of linages, strongly suggests that the evolution of specific traits has predisposed some lineages to novel threats. An understanding of: (i) the traits of plant species and how these traits mediate impacts of threats; and (ii) the distribution of species and how this influences exposure to threats and adequacy of specimen collection effort, offers a potential way forward for an initial assessment of which of the 1819 data-deficient taxa may be most at risk of extinction.

**Author Contributions:** Conceptualisation, C.R.G., M.B. and C.J.Y.; methodology, C.R.G.; software, C.R.G. and J.M.P.-B.; validation, C.R.G., T.M.L. and J.M.P.-B.; formal analysis, C.R.G. and J.M.P.-B.; investigation, C.R.G.; resources, C.R.G., T.M.L. and J.M.P.-B. data curation, C.R.G. and J.M.P.-B.; writing—original draft preparation, C.R.G., C.J.Y., J.M.P.-B. and M.B.; writing—review and editing, C.R.G., C.J.Y., M.B. and T.M.L.; visualization, C.R.G.; supervision, C.R.G.; project administration, C.R.G.; funding acquisition, not applicable. All authors have read and agreed to the published version of the manuscript.

**Funding:** This research received no external funding.

**Institutional Review Board Statement:** Not applicable.

**Data Availability Statement:** All data used in this study are available through the sources cited [37–39]. Location data for conservation-listed flora held in TPFL [38] are sensitive, and distribution upon reasonable request is regulated under licensed conditions.

**Acknowledgments:** We thank David Coates for providing suggestions on the content of this manuscript and staff in the DBCA Species and Communities programme for access to TPFL.

**Conflicts of Interest:** The authors declare no conflict of interest.

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
