# Peer review of "Distribution, Biogeography and Characteristics of the Threatened and Data-Deficient Flora in the Southwest Australian Floristic Region"

_diversity, doi:10.3390/d14060493_

Round 1
Reviewer 1 Report
I found this paper, in which a spatial analysis was performed by taking herbarium records of specimen locations and placing them into a GIS for an analysis of various categories of vegetation species conservation threat in the Southwest Australian Floristic Region, to be quite interesting. I believe this is a valuable contribution to the conservation literature for the area that can provide some guidance for where to target on-the-ground sampling efforts for conservation efforts. The author's do an excellent job of providing background to provide a basis for their work and to support their findings. In particular, the integration of supporting literature into the discussion is superb. My only notable concern regarding the paper is that there is no discussion of sampling bias in the methods section. Given that herbaria collections are not typically generated in a fashion to specifically support spatial conservation analysis, it seems that this would deserve being addressed by several sentences to fully caveat a potential issue in the analysis. To the author's credit, there is a sentence in the discussion that alludes to this, though in a more specific context. I have very few comments beyond this, and have submitted a PDF with a few wording suggestions that the authors may choose to integrate or not; I don't consider them mandatory. Also, I tend to like the inclusion of the "Oxford comma", so have included this on occasion, but leave it to the authors and editor as to whether they wish for this to be retained.
Overall, fine job and thank you for your contribution to the field.

Reviewer 2 Report
I have gone through the manuscript 1749137 “Distribution, biogeography, and characteristics of the threatened and data deficient flora in the Southwest Australian Floristic Region” and found it interesting for publication. However, authors need to improve their paper reasonably. The manuscript suffers certain limits because it represents a compilation of data without many critical arguments with regard to the plants used in Southwest Australian Floristic Region. Looking at Pubmed, I found many papers concerning diversity and threatened aspects of plants, still, I feel this manuscript has certain potential which needs to exploit through the publication of this manuscript. I suggest the following incorporations:
• The authors should first of all point out the novelty of these results with respect to those in the literature or at least make a comparative analysis, some of them are presented here. Were some species rare? Are the results evidencing that in the study area, the local flora is facing serious challenges due to over-harvesting and heavy grazing activities? Were some species linked to globalization and modernization rather than tradition?
• Authority of the genera in the running text should be mentioned, which is missing
• Updated references should be included in the article for Southwest Australian Floristic Region.
• Sources of data collection should be clearly mentioned
• Is there any impact of climate change on the endemic species in the study area, should be included in the discussion which should be supported by the latest references.
• Patterns of the invasiveness among the flora of SWAFR should be included in the discussion.
I recommend it for revision.
Reviewer 3 Report
The manuscript is of certain concern, and can be published after appropriate revision. In general, the manuscript has descriptive character, and the Introduction is of national, instead of international, scale. In addition, through the text, some inconsistencies and mistakes in English were found; I recommend double-checking the text in this term. I suggest to revise the manuscript based on my following comments and suggestions:
The abstract suffers from the lack of numeric characteristics of the obtained results. One of the examples is in line 18: “The majority of populations...”. Here, what percent is meant under “majority”? This comment should be applied to other similar cases in the abstract.
The list of key words appropriately reflects the content and the main terms of the manuscript.
Now, the section Introduction is of regional level, since it is focused on the study area only, i.e. the Southwest Australian Floristic Region. Actually, this is a research background description. However, I strongly suggest to highlight the international relevance of studies of threatened species in various regions of the world. I may advise the use of studies on IUCN assessments (e.g. https://dx.doi.org/10.24189/ncr.2020.033, https://doi.org/10.1016/j.jnc.2014.06.004, or others), biodiversity (e.g. https://dx.doi.org/10.24189/ncr.2020.061, https://doi.org/10.1016/j.jnc.2010.01.002, https://dx.doi.org/10.24189/ncr.2020.053, https://doi.org/10.1023/A:1016049605021, or others) of threatened plant species, as well as the well-known problem of plant species extinction (e.g. https://dx.doi.org/10.1186/s12870-020-02646-3, https://dx.doi.org/10.1111/ddi.12665, https://dx.doi.org/10.1002/ppp3.10146, or others) with a special focus on Mediterranean-climate regions (e.g. http://dx.doi.org/10.1016/j.cub.2019.07.063).
After this paragraph highlighting a relevance of the research topic, it is more suitable to pay attention to the background of the conducted research.
In Fig. 1 the category of “Priority 4 flora” is presented. But before this illustration, the authors should indicate in the main text, what does mean this category, if it is compared with “Data deficient flora” and “threatened flora”.
The paragraph in lines 92-95 should be moved to Discussion or in Conclusion. Research implications should be presented after your results and their comprehensive discussion.
Line 103: “limited spatial reliability of some records”. What does it mean? If the location of these records is not reliable, they should be removed from the analysis of spatial distribution of the flora. What percent was found of these unreliable records from the total number?
Lines 135-136: IUCN criteria and guidelines for assessment must be based ONLY on IUCN Red List Assessment Guidelines of the last versions for global and regional (sub-global) application! See them here among resources https://www.iucnredlist.org/. All other sources are secondary ones in comparison with the sources mentioned by me.
In general, Results are clearly presented. But I have some comments and suggestions.
In legends to sub-illustrations within Fig. 1 and Fig. 2, there are not indication of units for pixel colour. For instance, in Fig. 2c, >3, 3, 2, 1, but what units are mentioned? I guess, these are occurrences, if the authors say about data. I suggest to add indication of units in legends to avoid misunderstanding.
Fig. 3 (and other figures and tables) should be placed immediately after their citation in the text.
The section Discussion has the same problems, like it was found in Introduction. I mean the lack of comprehensive comparison and interpretation of the obtained results with the published data obtained in other regions of the world, including both tropical and temperate regions. Independently of climate type, the conservation patterns of the flora are generally similar in all countries at present, which is caused by anthropogenic influence of the vegetation cover, as well as climate changes (predominantly caused by human activities, too). I mean that the obtained results should be explained and interpreted in light of international (not only national) literature, as it is predominantly present now. In this regard, some paragraphs in Discussion have no enough references, like in sub-section 4.1, where again there is no comparison of the obtained results with situations in other regions of the world.
In Fig. 4 (it should be also placed a couple paragraphs above), there is a need to add the legend to colours of dendrogram.
Other parts of the Discussion needs to be updated on the basis of my comment on the literature (see above).
Finally, I think that there is a lack of the section Conclusion. In such a research paper, like this manuscript, the main conclusions and research implications should be concentrated in a certain section, namely Conclusion.
Round 2
Reviewer 3 Report
I appreciate the authors for adding sufficient corrections and providing replies to my comments and suggestions. All my suggestions were taken into account; otherwise, I have received response why the corrections cannot be accepted. In brief, I am satisfied with the updated version of the manuscript. I think that it is acceptable in present form.